# High-Temperature Oxidation Resistance of PDC Coatings in Synthetic Air and Water Vapor Atmospheres

**DOI:** 10.3390/molecules26082388

**Published:** 2021-04-20

**Authors:** Milan Parchovianský, Ivana Parchovianská, Peter Švančárek, David Medveď, Mateus Lenz-Leite, Günter Motz, Dušan Galusek

**Affiliations:** 1Centre for Functional and Surface Functionalised Glass, Alexander Dubček University of Trenčín, Študentská 2, 911 50 Trenčín, Slovakia; ivana.parchovianska@tnuni.sk (I.P.); dusan.galusek@tnuni.sk (D.G.); 2Joint Glass Centre of the Institute of Inorganic Chemistry Slovak Academy of Sciences, Alexander Dubček University of Trenčín and Faculty of Chemical and Food Technology Slovak University of Technology, Študentská 2, 911 50 Trenčín, Slovakia; peter.svancarek@tnuni.sk; 3Institute of Materials Research, Slovak Academy of Sciences, Watsonova 47, SK-043 53 Košice, Slovakia; dmedved@saske.sk; 4Ceramic Materials Engineering (CME), University of Bayreuth, D-95440 Bayreuth, Germany; Mateus.Lenz-Leite@uni-bayreuth.de (M.L.-L.); guenter.motz@uni-bayreuth.de (G.M.)

**Keywords:** PDC coatings, oxidation, synthetic air, water vapor

## Abstract

This work is aimed at the development and investigation of the oxidation behavior of ferritic stainless-steel grade AISI 441 and polymer-derived ceramic (PDC) protective coatings. Double-layer coatings of a PDC bond coat below a PDC top coat with glass and ceramic passive fillers’ oxidative resistance were studied at temperatures up to 1000 °C in a flow-through atmosphere of synthetic air and in air saturated with water vapor. Investigation of the oxide products formed at the surface of the samples in synthetic air and water vapor atmospheres, at different temperatures (900, 950, 1000 °C) and exposure times (24, 96 h) was carried out on both uncoated steel and steel coated with selected coatings by scanning electron microscopy (SEM) and X-Ray diffraction (XRD). The Fe, Cr_2_O_3_, TiO_2_, and spinel (Mn,Cr)_3_O_4_ phases were identified by XRD on oxidized steel substrates in both atmospheres. In the cases of the coated samples, m- ZrO_2_, c- ZrO_2_, YAG, and crystalline phases (Ba(AlSiO_4_)_2_–hexacelsian, celsian) were identified. Scratch tests performed on both coating compositions revealed strong adhesion after pyrolysis as well as after oxidation tests in both atmospheres. After testing in the water vapor atmosphere, Cr ions diffused through the bond coat, but no delamination of the coatings was observed.

## 1. Introduction

When designing parts and products, along with their structural characteristics (such as strength), it is also necessary to consider the impact of environmental aggressiveness on their functionality and required durability. If it is determined that the environmental effects on the performance of the part is too much, designing protection can eliminate or suppress the adverse impact. An optimal design ensures long-term protection with minimal implementation and maintenance costs over its required service life.

A simple solution to protect stainless steel from oxidation at elevated temperatures, and significantly extend its operation lifetime, is the application of suitable coatings, which are resistant to different oxidation atmospheres at higher temperatures [1]. Currently, there are several common methods for the preparation of ceramic coatings, including thermal spraying, chemical vapor deposition (CVD) [2], physical vapor deposition (PVD) [3], and sol–gel techniques [4,5]. Some disadvantages of these proven methods are a high degree of porosity, insufficient adhesion of prepared coatings, and they usually use expensive specialized equipment (e.g., plasma and vacuum technology). A suitable alternative is the development of protective PDC coating systems [6] by the thermal decomposition (pyrolysis) of organoelemental compounds (precursors), such as polysiloxanes [7], polycarbosilanes [8], or polysilazanes [9]. The preparation of ceramic components from organosilicon polymer precursors usually involves the following three steps: (1) selection or synthesis of an organosilicon polymer, with a suitable composition, structure, and a controlled molecular weight distribution, (2) forming, followed by thermally or catalytically induced crosslinking of the polymer precursor in the temperature range of 100–300 °C to an organic/inorganic system, (3) pyrolytic transformation of a crosslinked system in an inert or reactive atmosphere, at 500–1500 °C, to the desired amorphous/crystalline ceramic [10,11]. These proposed methods of preparation allow the application of coating technologies, such as spraying [12] and dipping [13] in a liquid precursor, doctor blade method [14], or tape casting [15] followed by pyrolytic conversion of a liquid precursor to an amorphous/crystalline ceramic. The advantages of precursor methods are their high chemical homogeneity of the products, easy formability, and their ability to prepare non-oxide ceramic materials in the form of thin layers [16,17], protective coatings [18,19], membranes [20], or fibers [21]. The disadvantages are high shrinkage, technological difficulties associated with the sensitivity of organosilicon polymers to oxygen and humidity, and the problematic preparation of bulk materials due to the intensive evolution of gases (hydrogen, methane, hydrocarbon residues) during pyrolysis [22]. As mentioned above, the conversion of the polymer to the ceramic is accompanied by a release of gaseous by-products, which cause weight loss and overall shrinkage of the final ceramic layer (up to 50 vol %, depending on the precursor), which usually leads to the formation of defects, such as cracks and pores, or delamination of the layer, either partially or fully [23]. The change in volume during conversion of the precursor can be compensated by an admixture of active (ZrSi_2_ [24] or TiSi_2_ [25]) and passive fillers (BN [26], ZrO_2_ [27], or Al_2_O_3_ [28]), with suitable laboratory melted or commercial oxide glasses. The active/passive fillers are mainly responsible for suppressing shrinkage, while the glasses are responsible for the densification and sealing of the coating system. However, the application temperature of the coatings is usually limited by the melting temperature of the glass frit. Therefore, suitable glasses must be selected and used.

To ensure the required adhesion of a coating to metallic substrates, duplex-type coating systems, which consist of a bond coat and a ceramic top coat, are frequently used. The role of the bond coat is to protect the stainless steel and improve the bonding between the ceramic top coat and steel substrate. The top coat supports the protection of the base layer from oxidative and corrosive attacks and exhibits phase stability during long-term, high-temperature exposure and thermal cycling. Hence, it is possible to reduce the temperature of the metallic substrates, thereby increasing the lifetime of the metal. The oxidation and corrosion resistance of PDC coatings have been studied in several works [29,30,31]. In our previous studies [32], we developed double-layer PDC coatings for steel substrates with thermal stability up to 950 °C using passive fillers. Testing these coatings in an oxidative atmosphere (ambient air) at 1000 °C revealed their unsatisfactory oxidation protection behavior. Although these coatings all fail when exposed to 1000 °C and still face many challenges, their protective effect was demonstrated up to 950 °C.

The present work deals with the preparation, characterization, and study of the oxidation behavior of duplex-type composite PDC coating systems, which consist of a bond coat and a ceramic top coat. The stainless steel and prepared coatings were tested in an atmosphere of air and water vapor at temperatures up to 1000 °C. The microstructure, phase composition, and oxidation resistance of the prepared coatings were studied. The assessment of oxidation resistance was carried out through detailed characterization of the coatings by observing weight changes, microstructure, adhesion, chemical and phase composition before and after oxidation tests, the formation of oxidation products, and delamination of the formed oxides or protective layers.

## 2. Materials and Methods

### 2.1. Preparation of the Coatings

A dual PDC coating system consisting of a bond coat, made using Durazane2250 (Merck KGaA, Darmstadt, Germany) as the precursor and a composite ceramic top layer prepared from fillers and an organic liquid polysilazane, Durazane1800 (Merck KGaA, Darmstadt, Germany) was developed. The process started with cutting and pre-treating the steel substrates and the preparation of the coating slurry. AISI441 ferritic stainless steel was used as the default metal substrate for double-layer coatings. At first, the stainless steel was cut into 10 × 15 × 1 mm sheets, ultrasonically cleaned in acetone, ethanol, and deionized water, and then dried. The corners and edges of the steel samples were chamfered with sandpaper. The bond coat was subsequently applied via dip-coating (dip-coater RDC 15, Relamatic, Glattbrugg, Switzerland) and fixed by pyrolysis (temperature −450 °C, holding time–1 h, furnace–Nabertherm^®^ N41/H, Nabertherm, Lilienthal, Germany) before the composite top coat was applied by spray-coating (Nordson EFD 781S-SS) and again pyrolyzed in air. The application of the top layer was done to both sides of the steel substrate. First, the stainless steel was sprayed from one side; then, the samples were dried in an electric oven for 24 h at 120 °C. After drying, the second side of the stainless steel was sprayed under the same conditions. Two compositions of the top coat were prepared and denoted as C2c and D4. The volume fractions of the individual coating components are given in Table 1. Suspensions of both compositions consisted of di-n-butyl ether (Acros Organics BVBA, Antwerp, Belgium), dispersant (Disperbyk 2070, BYK-Chemie GmbH, Wesel, Germany), liquid polysilazane Durazane1800, ZrO_2_ stabilized with 8 mol % Y_2_O_3_ (8YSZ, Inframat, Manchester, CT, USA), and commercially available glass frits (barium silicate glass–G018-281, Schott AG, Mainz, Germany). In the case of the D4 suspension, a polycrystalline Al_2_O_3_-Y_2_O_3_-ZrO_2_ (AYZ) precursor powder prepared by the modified Pechini sol–gel method was used as an additional filler to minimize volume changes during the transformation of the precursor to ceramic. The preparation and basic characterization of AYZ precursor powder was described in our previous work [18]. The basic properties of the filler materials are listed in Table 2. The final pyrolysis of the ceramic coatings was performed in air (Nabertherm^®^ N41/H, Nabertherm, Lilienthal, Germany) at 850 °C for 1 h with a heating rate of 3 °C/min.

### 2.2. Oxidation Tests

To assess the protective ability of the prepared coatings, static oxidation tests were performed while the resistance of uncoated steel was also monitored as a reference material. The coated samples and uncoated steel substrates were placed in a horizontal tube furnace (Clasic 0213T, Praha, Czech Republic) and heated to a pre-set temperature with a heating rate of 5 °C/min in a flowing atmosphere. The samples were exposed to synthetic air (purity 99.5%, Siad, Bratislava, Slovakia) and air saturated with water vapor (equilibrium temperature of 80 °C) at temperatures of 900, 950, and 1000 °C, and residence times of 24 h and 96 h.

### 2.3. Characterization Methods

After each test, coated and uncoated samples were weighed, and their weight changes were determined to assess the protective function of the prepared coatings. The adhesion of the coating to the substrate was determined the by scratch test method with progressively increasing load. The scratch tests of coated samples, before and after oxidation tests, were carried out on universal tribometer (Bruker UMT 3, Bruker, Billerica, MA, USA) with a progressive loading from 2 to 20 N. The microstructure and chemical composition of the coatings, before and after the oxidation tests, were examined on a scanning electron microscope (JEOL JSM 7600 F, JEOL, Tokyo, Japan) equipped with an EDX detector. X-ray powder diffraction (PANalytical Empyrean DY1098 (Panalytical, BV, Almelo, The Netherlands)) with a Cu anode and X-ray wavelength λ = 1.5405 Å was used in the range 10–80° 2θ to determine the phase composition of the stainless steel and prepared coatings before and after oxidation tests.

## 3. Results and Discussion

### 3.1. Weight Changes Measurement

The oxidation of metals is a complex process, during which partial reactions take place at the phase interfaces of the metal–oxide–oxygen system and in the layer of oxides formed in the process. Weight changes of uncoated steel and steel coated with C2c and D4 compositions are shown in Table 3 with differing of oxidation times, temperatures, and atmospheres.

At all tested times and temperatures, there were significant weight gains on uncoated steel substrates due to the formation of oxide products on their surface. The weight gain of steel after oxidation tests increased with increasing temperature. A more detailed study of steel oxidation is described in our previous works [30,32]. Slight weight losses were detected for the C2c and D4 coated steels when tested in synthetic air at 900 °C. However, at 950 °C and 1000 °C, a significant weight loss was observed in composition C2c due to extensive delamination and coating failure. On the other side, composition D4 showed small weight gains at 950 °C and 1000 °C, whereas the uncoated substrates were already strongly oxidized at these temperatures. The weight gain in the D4 coating after 96 h and at 1000 °C was due to the formation of an oxide layer that was 77% lower than that for uncoated steel tested at the same temperature and exposure time. These results indicate that the D4 coating exhibited better protective ability in synthetic air than the C2c coating.

The atmosphere of water vapor was much more aggressive, which also resulted in higher weight gains of steel and coated samples compared to samples exposed to synthetic air. The uncoated steel exposed to water vapor at 1000 °C for 96 h achieved a weight gain of 15.46 mg/cm^2^. This is one order of magnitude higher than the weight gain of 1.25 mg/cm^2^ measured for the steel exposed to synthetic air under the same conditions. As for the C2c and D4 compositions exposed to water vapor, only slight weight losses (0.24 and 0.20 mg/cm^2^ for C2c and 0.11 and 0.14 mg/cm^2^ for D4) at shorter oxidation times (24 h) and lower temperatures (900 and 950 °C) were observed. In the C2c-coated steel exposed to water vapor at 1000 °C, the weight gain after 96 h was comparable to that of the uncoated steel substrate (13.3 mg/cm^2^). Peeling of the layer from the steel substrate was not observed. Only a negligible weight gain of only 0.56 mg/cm^2^ was measured for the D4-coated steel, indicating better protective function compared to the coating C2c, even in the water vapor atmosphere. From the two tested coatings, the D4 coating exhibited the lowest porosity with pore sizes smaller than 10 µm, thus acting as the most robust oxidation protection system. Apart from the discernible porosity, no continuous cracks across the coatings were observed. With increasing time (96 h) and temperature (1000 °C), a gradual increase in the weight of the samples was observed due to the increase of the amount of oxide products on the steel substrate, as will be discussed in more detail in the section on SEM/EDXS analysis.

### 3.2. XRD Measurement

The surfaces of the investigated samples were analyzed by XRD to identify phases formed before and after oxidation tests. XRD patterns of stainless steel and coated samples before and after 96 h of oxidation tests at 950 °C in atmospheres of air and water vapor are shown in Figure 1a–c. Only Fe (PDF-96-901-3474) was identified by XRD for the stainless steel before oxidation (Figure 1a). After oxidation tests, Fe, TiO_2_ (PDF-96-900-4143), Cr_2_O_3_ (96-901-4037), FeTiO_3_ (PDF-96-900-0908), SiO_2_ (PDF-96-900-1579), and spinel (Mn,Cr)_3_O_4_ (PDF-96-900-5292) were identified from XRD patterns of uncoated steel substrates exposed to synthetic air (Figure 1a). These data agree with our previous results, where we performed oxidation tests in an atmosphere of ambient air at temperatures of 900, 950, and 1000 °C [32]. In this work [32], stainless steel substrate was covered by the passivation Cr_2_O_3_ layer after oxidation, with a TiO_2_ and a (Mn,Cr)_3_O_4_ spinel non-protective upper layer, while no Fe_2_O_3_ was detected by XRD and EDXS. Since ferritic stainless steel usually contains a small amount of Mn, a spinel (Mn,Cr)_3_O_4_ layer formed on top of the compact, and a protective Cr_2_O_3_ layer at temperatures above 800 °C [33]. The formation of the (Mn,Cr)_3_O_4_ spinel on top of the Cr_2_O_3_ scale can be attributed to the diffusion of Mn ions through the Cr_2_O_3_. The (Mn,Cr)_3_O_4_ spinel is formed due to the strong affinity of Mn to O_2_ and its high diffusion coefficient (D), which is higher than those of Fe or Cr: D_Mn_ > D_Fe_ > D_Cr_ [34]. The (Mn,Cr)_3_O_4_ spinel is predominantly formed as the outer layer; thus, it can reduce the chromium activity and stabilize the oxidation rate. The oxidation of uncoated steel in water vapor at temperatures up to 1000 °C resembles the oxidation in synthetic air. The oxide layer has a similar chemical composition. However, the FeTiO_3_ and SiO_2_ phases were not identified by XRD. Exposure of Fe-Cr alloys to an oxidizing atmosphere containing water vapor at temperatures above 500 °C usually results in degradation of the passivation oxide layer of Cr_2_O_3_ due to the release of volatile compounds (CrO_3_ and CrO_2_(OH)_2_) according to the following reactions (1, 2, 3) [35]:½ Cr_2_O_3_ (s) + ¾ O_2_ (g) → CrO_3_ (g)(1)
½ Cr_2_O_3_ (s) + H_2_O (g) + ¾ O_2_ (g) → CrO_2_(OH)_2_ (g)(2)
½ Cr_2_O_3_ (s) + ½ H_2_O (g) + ½ O_2_ (g) → CrO_2_(OH) (g)(3)

The evaporation rate of Cr_2_O_3_ is linearly dependent on the time of exposure in an oxidizing atmosphere, and higher temperatures lead to higher evaporation rates [36]. The evaporation of Cr leads to its loss from the protective passivation layer, which results in the formation of an unprotected iron-containing oxide layer and a subsequent rapid increase in the oxidation rate [37]. This is consistent with the results of the weight gain measurements in this work, which indicate a significant increase in the oxidation rate after oxidation tests at 1000 °C after 96 h.

In the coated samples, XRD was used to analyze the top layer before and after oxidation tests. The oxidation products of steel could only be detected if the coating peeled off or if the oxidation products penetrated or diffused to the coating surface. In both tested coatings (C2c–Figure 1b), (D4–Figure 1c), the predominant phases are monoclinic (PDF-96-901-6715) and cubic (PDF-96-210-1235) ZrO_2_. A peak belonging to quartz SiO_2_ (PDF-96-901-2602) was also observed, which was probably formed by crystallization of the glass frit during pyrolysis. After 96 h of oxidation tests in both atmospheres at 950 °C, monoclinic and cubic ZrO_2_ were again detected as the main phases. The presence of SiO_2_ was only confirmed after tests in synthetic air, while no SiO_2_ was detected after tests in water vapor. The crystallization of glass fillers also resulted in the formation of other crystalline phases, such as Ba(AlSiO_4_)_2_ (hexacelsian (PDF-01-088-1049) and celsian (PDF-96-201-3138)). The formation of these phases was facilitated by the relatively low glass transition temperatures and crystallization of the commercial glasses used as fillers. Hexacelsian Ba(AlSiO_4_)_2_, a polymorph of celsian, precipitated as the main crystalline phase during the oxidation tests. In the case of the D4 coating (Figure 1c), yttrium-aluminum garnet (YAG, PDF-96-431-2143) was also identified, which was already detected in the XRD pattern of the AYZ precursor powder after calcination.

The absence of peaks belonging to SiO_2_ in coatings tested in water vapor can be explained by the incorporation of SiO_2_ into other crystalline/amorphous phases or by reaction of SiO_2_ with water vapor and the subsequent release of volatile products, which is consistent with the measured weight loss values at shorter oxidation times. It has been reported that SiO_2_ can form volatile products (hydroxides or oxyhydroxides) in atmospheric pressure environments containing water vapor [38]. The possible reaction of SiO_2_ with water vapor can be described by the following reactions (Equations (4)–(6)) [38]:SiO_2_ (s) + H_2_O (g) → SiO(OH)_2_ (g)(4)
SiO_2_ (s) + 2 H_2_O (g) → Si(OH)_4_ (g)(5)
2 SiO_2_ (s) + 3 H_2_O (g) → Si_2_O(OH)_6_ (g)(6)

According to these reactions, the species SiO(OH)_2_ and Si(OH)_4_ are the predominant volatile species of the reactions in the water vapor atmosphere. It has been reported that for processes at 0.1 MPa and temperatures below ≈1300 °C, SiO_2_ volatility can be attributed to the reaction shown in Equation (5) [38]. The volatility of SiO_2_ in the water vapor atmosphere also depends on the temperature and partial pressure of the water vapor. The evaporation of SiO_2_ is promoted in an atmosphere with a high water vapor concentration [39].

After 96 h of oxidation at 950 °C, the presence of oxidation products ((Mn,Cr)_3_O_4_ and Cr_2_O_3_) arising from oxidation of the substrate can be observed in the XRD patterns of the C2c coating (Figure 1b) in addition to the phases present in the coating residue. This indicates the steel substrate was attacked by the corrosive medium. No crystalline oxidation products were detected by XRD in the D4 layer (Figure 1c).

### 3.3. Adhesion Strength

To ensure oxidation resistance, the PDC coatings must have good adhesion to the metal substrate. Coatings with weak adhesion to stainless steel may delaminate during oxidation and thus lose their protective function. Scratch tests were performed to obtain the critical load for evaluating the quality of coatings. Figure 2 shows the dependence of the acoustic emission signals and frictional force on the normal load after scratch testing of both the pyrolyzed and oxidized samples. The critical loads at which the acoustic emission signal rapidly increases are indicated by arrows in the graphs (Figure 2). The results indicate that both coatings adhere well to stainless steel substrate after pyrolysis. The first adhesive–cohesive failure of the as-pyrolyzed D4 coating, which was determined by the first significant increase in acoustic emission, occurred at a normal load of Fn ≈ 17 N. The adhesion decreased after the oxidation tests for both the C2c and D4 coatings. After oxidation tests in both synthetic air and water vapor, a larger ripple in the acoustic emission profile was visible for the C2c coated system. The adhesion was slightly lower, and the first layer failure was recorded at Fn ≈ 14 N. By default, the critical adhesion force required to break the coating and tear it from the substrate, which is determined from the scratch morphology, is also used to assess the adhesion. However, the critical force, Fc, at which the first significant damage to the coating occurs under normal load and is recorded from the acoustic emission signal could not be detected in the scratch morphology. Therefore, the exact location where the first layer failure occurred could not be determined.

The morphology of scratches in the coatings C2c and D4 after pyrolysis at 850 °C for 1 h is shown in Figure 3. In all cases, plastic deformation and apparent compaction of the layer at the scratch site were observed as a result of the gradual penetration and increasing load of the indenter, which was most likely related to the higher porosity in the coatings. Porosity, and large pores formed after oxidation tests in particular, are likely to decrease the adhesion strength. The pores that occur between larger filling particles may lead to the coating’s cohesive failure.

### 3.4. SEM Investigation

SEM/EDXS analysis was used for a detailed study of the surfaces and cross-sections of uncoated and coated samples before and after oxidation tests. The SEM micrographs of surfaces and cross-sections of the C2c and D4 coated steel samples before oxidation tests are shown in Figure 4. After pyrolysis in air at 850 °C for 1 h, homogeneous and almost dense protective coatings, with only small pores size, were prepared. Both coatings, C2c and D4 (Figure 4), showed no significant cracks at the surface or in the cross-sections. The coatings contain some residual porosity, but the pores were closed.

In addition, an EDXS mapping was performed on the cross-section of the D4 coating (Figure 5). The EDXS maps (Figure 5) show that the main component of the bond coat after pyrolysis was Si. Other elements, i.e., Zr, Si, Ba, Al, and O, were identified in the top coat. The presence of Fe and Cr is clearly seen in the stainless steel substrate. No oxidation products were observed at the stainless steel/bond coat or bond coat/top coat interfaces after pyrolysis.

SEM micrographs showing the surface of stainless steel after oxidation tests in both atmospheres at different temperatures (900, 950, 1000 °C) and times (24, 96 h) are shown in Figure 6. At the test temperatures, the steel underwent significant oxidation, with a thick layer of oxidation products, (Mn,Cr)_3_O_4_ and Cr_2_O_3_ in particular, formed on the surface of the material, as confirmed by EDXS. Different temperatures, times, and atmospheres markedly influenced the structure and morphology of the oxidized surface and the size of the newly formed crystals phases (Figure 6). After exposure to a lower temperature (900 °C) and shorter oxidation time (24 h), the surface of the steel was uniform and homogeneous, and it was covered with fine crystals. After exposure at a higher temperature (1000 °C) and longer oxidation time (96 h), the oxide layer was coarser. Then, the weight gains measured after the oxidation tests can be attributed to an increase in the amount of oxides formed on the surface of the samples, which increase with temperature and time.

Figure 7 shows the SEM micrographs of the selected coatings after oxidation tests in synthetic air. Compared to the coatings after pyrolysis (Figure 4), a significant increase in the porosity of the layers and in the pores size can be observed. The dark gray areas in the cross-sectional SEM images, indicated by the yellow arrows, were identified by EDXS analysis as residual porosity, with the pores filled with synthetic resin used for embedding of samples during the preparation of cross-sections. The expansion of the existing pores is attributed to used glass frits softening and the subsequent differential sintering of individual components in the layers by viscous flow and complemented by crystallization of phases from residual glass (celsian, hexacelsian). At 900 °C (Figure 7), the C2c coating showed good adhesion to the metal substrate, with the occasional occurrence of cracks at the metal–coating interface. However, further expansion of the pores and increased frequency of cracking between the steel substrate and the coating was observed at higher temperatures, which eventually led to a gradual delamination of the coatings.

As already confirmed by visual inspection of the coatings and the weight loss measurements, exposure of the samples to 1000 °C (Figure 7) led to full-area delamination and loss of oxidation resistance. More promising results were observed for the composition D4, which contains the AYZ powder as passive filler. At all tested temperatures, the D4 coating showed lower porosity than the C2c coating. SEM cross-sectional images (Figure 7) also showed good coating adhesion, even at higher test temperatures and after longer exposure times. The cross-sections of these coatings (Figure 7) also show the absence of cracks that penetrate from the surface to the metal substrate. The better oxidation resistance of the D4 precursor powder layers can be attributed to the different microstructure of the coatings. The increased occurrence of pores and cracks in the C2c coating allowed a faster penetration of synthetic air through the coating to the metal substrate, which caused a faster growth of the oxide layer at the metal/coating interface, causing subsequent delamination and disintegration of the layer. Based on the evidence shown above, it is proposed that the addition of the AYZ powder precursor to the composition D4 helped create a solid and fragmented structure. This allowed the decomposition gases to escape from the system during pyrolysis, and thus, it effectively reduced the size and number of pores. Furthermore, the addition of the AYZ powder significantly reduced differential sintering during the tests in synthetic air. Although the coatings are not fully dense, as is required for environmental barrier coatings, microstructures with such residual porosity are favorable for thermal stability of the coatings, because it contributes to the relief of residual stresses during heating and cooling cycles.

The oxidation process of the coated samples in a water vapor atmosphere differed from the process in synthetic air. SEM cross-sections of the C2c and D4 coatings after oxidation tests in water vapor after 96 h are shown in Figure 8.

According to visual inspection, the coatings appeared to adhere well after the tests, did not peel off, and no visible changes were observed on the surface. Partial corrosion protection was observed in a water vapor atmosphere only at 900 °C for the C2c coating. At higher temperatures (950 °C, 1000 °C), the top coat was disrupted due to increased porosity and pore formation caused by differential sintering of the individual coating particles. On the other hand, for both compositions, no delamination of the bond coat from the metal substrate was observed. In comparison, the C2c samples tested in synthetic air (Figure 7) showed local to total delamination of the coating, especially at higher temperatures and longer exposure times. Upon closer examination of the SEM cross-sections of the coatings, it is evident that the prepared D4 coatings are a more effective anti-corrosion barrier at the tested temperatures and times. Extensive layer disruption was not observed, even at high temperatures and long exposure times, showing less porosity than the C2c coating. However, an oxide layer can be observed at the bond coat/top coat interface for both compositions at 950 and 1000 °C. This suggests that during the tests in water vapor, Cr and Mn diffused through the bond coat, reacted with the atmosphere, and formed a layered oxide structure at the bond coat/top coat interface. No oxide scale was observed at the steel/bond coat interface. To confirm the assumption, SEM/EDXS analysis of the D4 composition tested in water vapor at 1000 °C for 96 h was performed (Figure 9).

The EDXS cross-section analysis (Figure 9) confirmed the presence of only Si and O in the bond coat, which was the result of conversion of the used polymer, Durazane2250, to SiO_2_ during pyrolysis in air. On the other hand, Al, Zr, Si, and Ba were identified in the upper glass-ceramic layer. EDXS cross-sectional analysis also revealed the presence of a small amount of Mn in the top coat, which likely diffused out of the steel during the formation of the Cr_2_O_3_ layer, which was located at the interface between the bond and top coat. Moreover, some Ba appeared in the steel substrate (Figure 9). This indicates the mobility of the Ba ions from the glass frit and the metal atoms of the steel and even the exchange of these ions at the coating/steel interface by interdiffusion.

## 4. Conclusions

The oxidation behavior of stainless steel and two protective composite coatings, C2c and D4, were investigated in synthetic air and water vapor atmospheres. The D4 coating exhibited better protective properties than the C2c coating in both atmospheres and at temperatures of 900 °C and 950°C. At 1000 °C, no significant oxidation damage was observed for the D4 coating, but the C2c coating showed gradual peeling of the protective layers and the loss of oxidation resistance. The water vapor atmosphere was more aggressive, which also resulted in higher weight gains of both the uncoated steel and the coated samples than in samples exposed to only the synthetic air atmosphere. The scratch tests of both compositions showed good adhesion of both as-pyrolyzed coatings and the coatings after oxidation tests in both atmospheres. The Fe, Cr_2_O_3_, TiO_2_, and spinel (Mn,Cr)_3_O_4_ phases were identified from XRD patterns of oxidized uncoated steel substrates in both atmospheres. In coated samples, m–ZrO_2_, c–ZrO_2_, YAG, and Ba(AlSiO_4_)_2_–hexacelsian, celsian were detected. In water vapor, Cr^3+^ ions diffused through the bond coat, forming a Cr_2_O_3_ layer at the bond coat/top coat interface, but no delamination was observed.

## Figures and Tables

**Figure 1 molecules-26-02388-f001:**
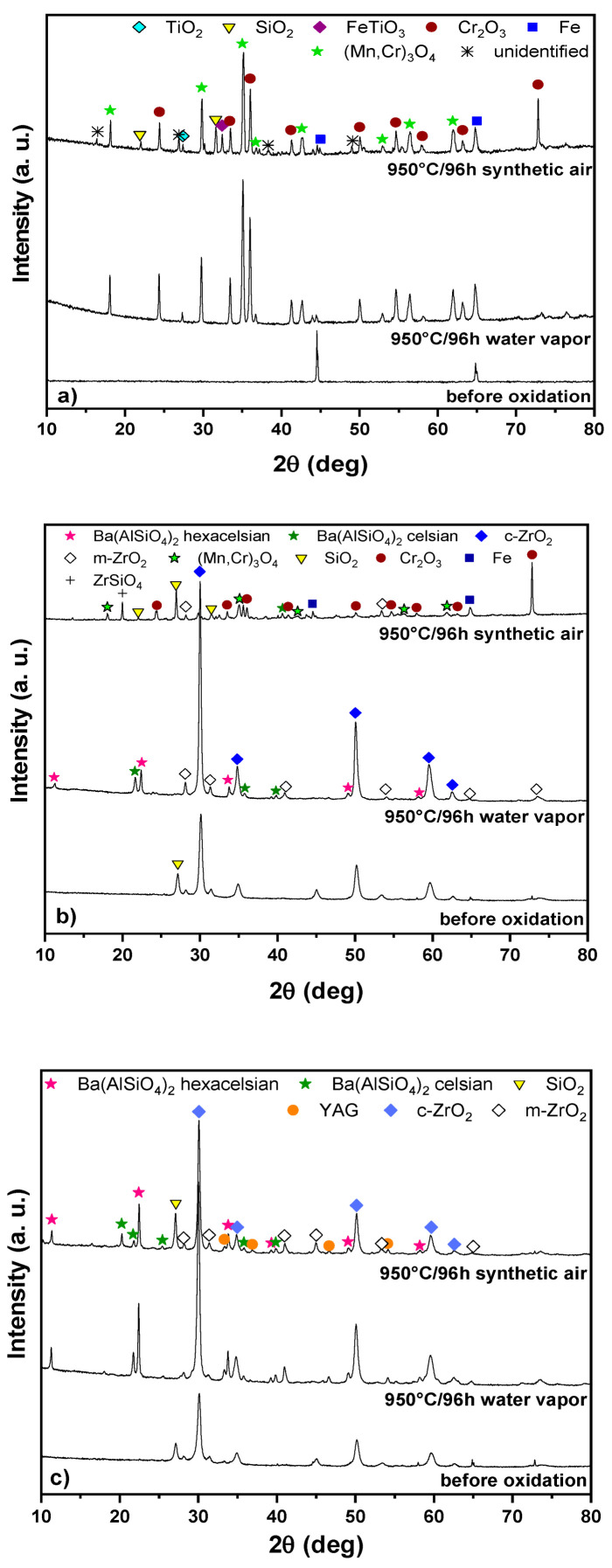
XRD diffraction patterns of tested materials before and after oxidation tests at temperature of 950 °C after 96 h exposure (**a**) steel, (**b**) C2c, (**c**) D4.

**Figure 2 molecules-26-02388-f002:**
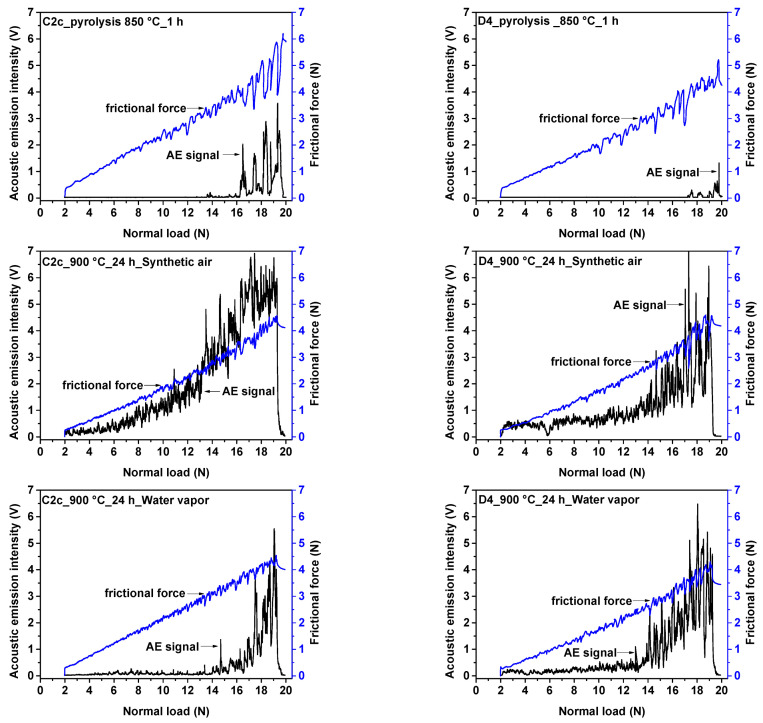
The dependence of acoustic emission signal and frictional force on the normal load during scratch tests of C2c and D4 coatings before and after oxidation tests (48 h at 900 °C).

**Figure 3 molecules-26-02388-f003:**
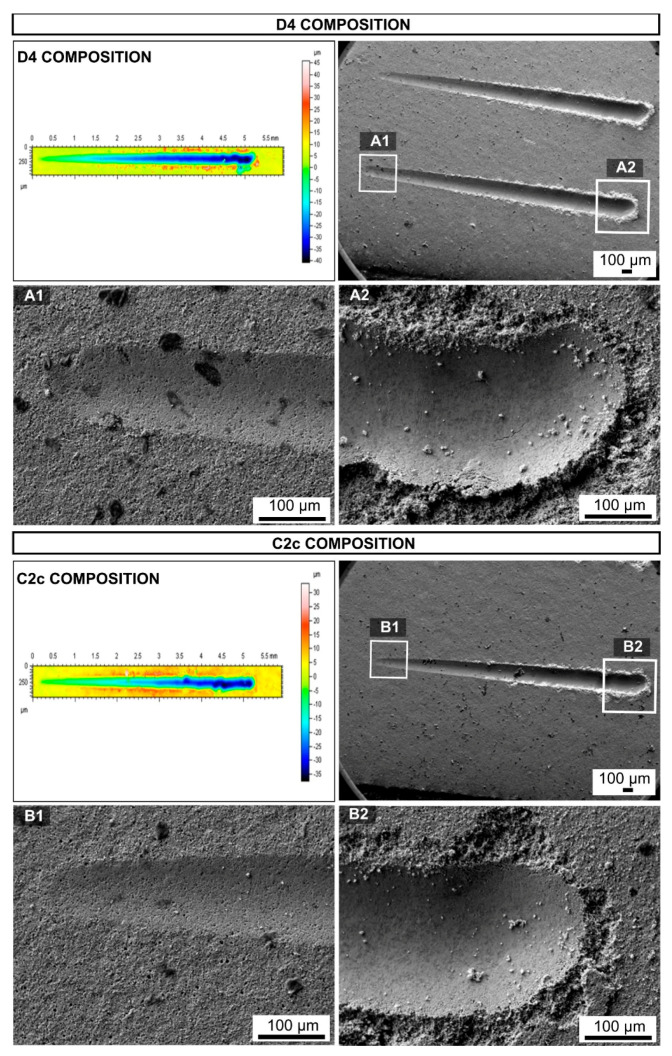
Scratch morphology in the coatings C2c and D4 after pyrolysis in ambient air at 850 °C/1 h.

**Figure 4 molecules-26-02388-f004:**
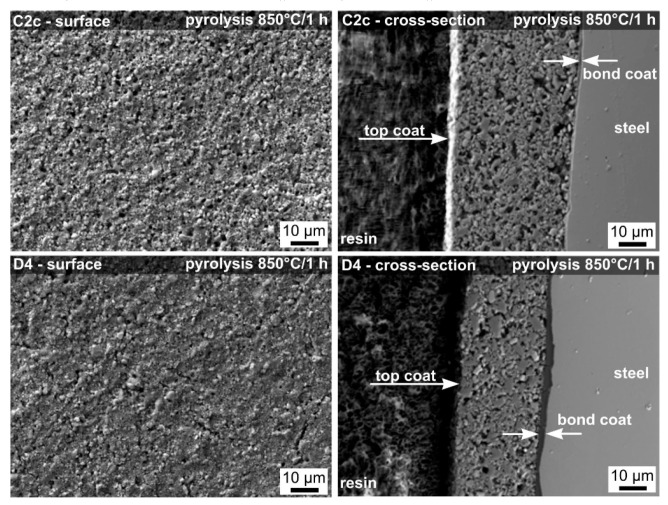
SEM surfaces and cross-sections of compositions C2c and D4 after pyrolysis at 850 °C.

**Figure 5 molecules-26-02388-f005:**
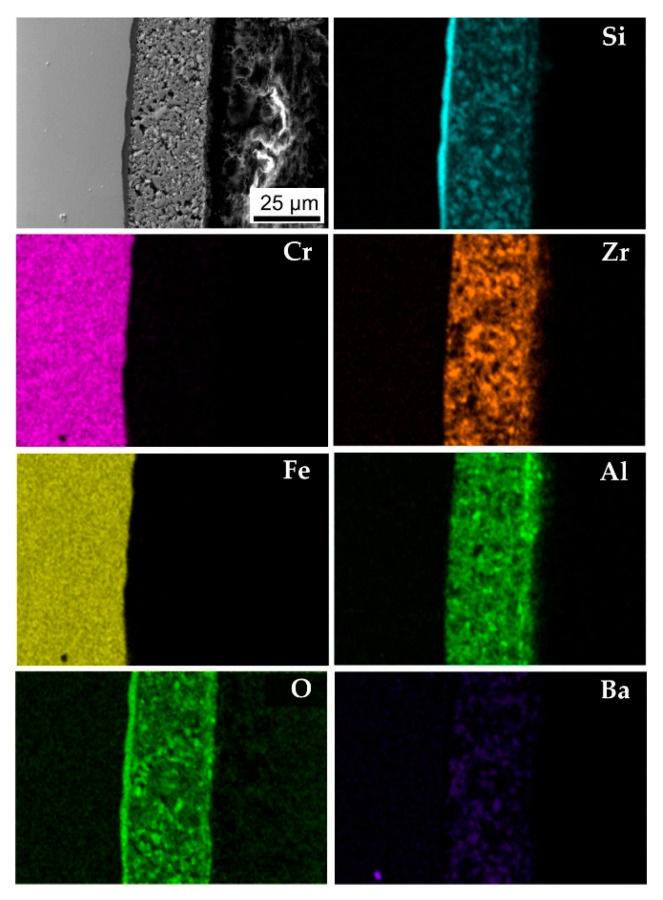
SEM/EDXS analysis of composition D4 after pyrolysis at 850 °C.

**Figure 6 molecules-26-02388-f006:**
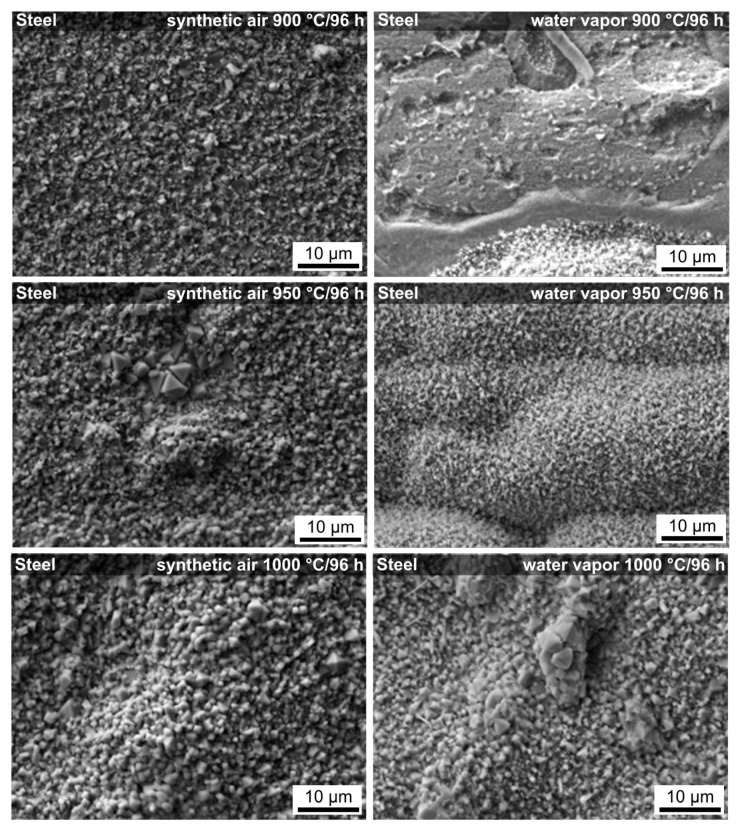
SEM micrographs of uncoated steel surfaces after oxidation tests in synthetic air and water vapor.

**Figure 7 molecules-26-02388-f007:**
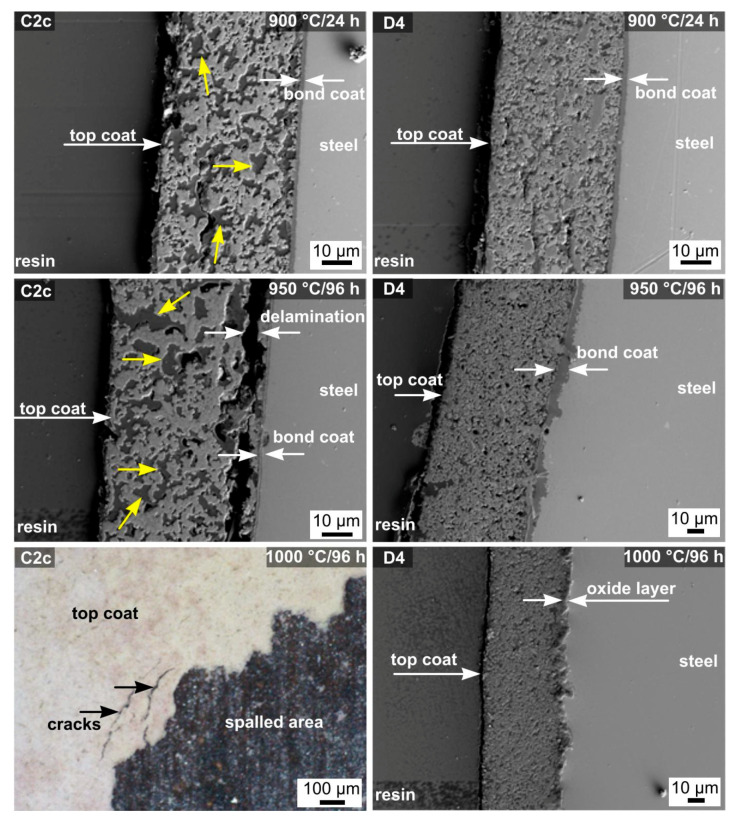
SEM cross-sections of C2c and D4 compositions after oxidation tests in a synthetic air atmosphere at different temperature and times. The yellow arrows show the pores filled with synthetic resin.

**Figure 8 molecules-26-02388-f008:**
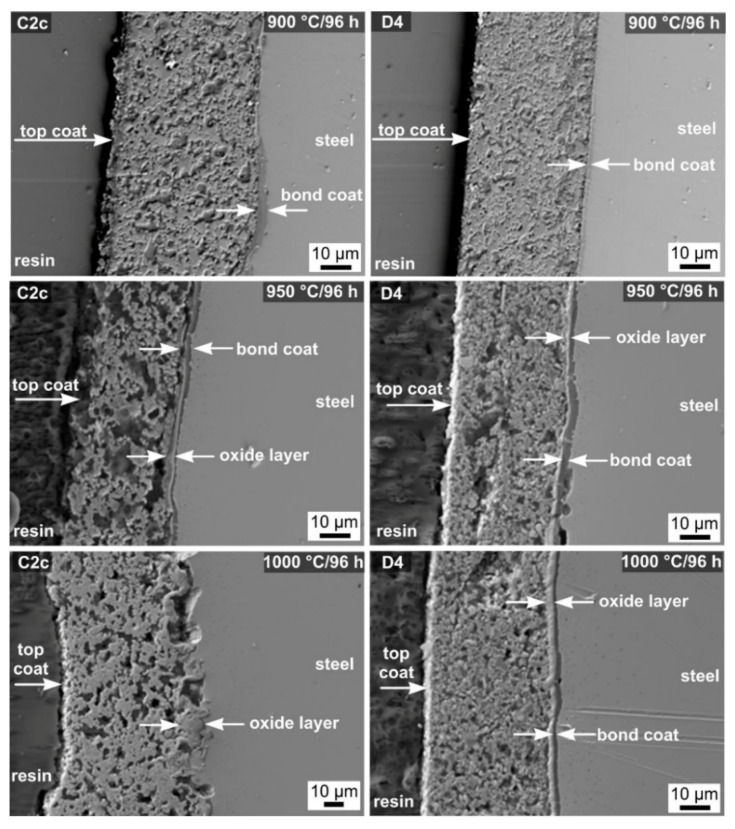
SEM cross-sections of layers C2c and D4 after oxidation tests in water.

**Figure 9 molecules-26-02388-f009:**
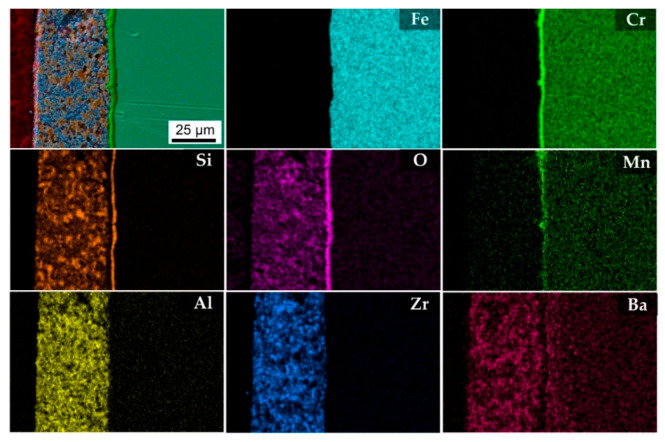
SEM/EDXS cross-sections of layer D4 after oxidation tests in water vapor at 1000 °C for 96 h.

**Table 1 molecules-26-02388-t001:** Compositions of prepared composite coatings (vol %) and their coefficient of thermal expansion (CTE, × 10^−6^ K^−1^).

Compositions	Durazane1800	YSZ	Glass G018-281	AYZ	CTE
**C2c**	30	35	35	-	10.1
**D4**	30	17.5	35	17.5	9.4

**Table 2 molecules-26-02388-t002:** Basic properties of filler materials.

Passive Fillers	d50 (µm)	ρ (g/cm^3^)	CTE (10^−6^/K)
8YSZ	0.5	6.1	11.5
AYZ	1–10	4.6	8.6
Glass G018-281	0.5–5	2.7	12.1

**Table 3 molecules-26-02388-t003:** Weight changes of test samples after oxidation tests in atmospheres of synthetic air or water vapor (mg/cm^2^).

		Synthetic Air			Water Vapor		
**Composition**		900 °C	950 °C	1000 °C	900 °C	950 °C	1000 °C
**AISI 441**	24 h	0.21	0.44	0.72	0.21	0.38	0.70
	96 h	0.45	0.77	1.21	0.48	0.80	15.46
**C2c**	24 h	−0.11	−1.33	−4.95	−0.24	−0.20	−0.13
	96 h	−0.14	−5.06	−5.48	−0.1	0.12	13.33
**D4**	24 h	−0.07	−0.03	0.03	−0.11	−0.14	−0.01
	96 h	0	0.15	0.27	−0.05	0	0.56

## Data Availability

The data presented in this study are available in this article.

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
