# Peer review of "High-Temperature Oxidation Resistance of PDC Coatings in Synthetic Air and Water Vapor Atmospheres"

_molecules, 2021, doi:10.3390/molecules26082388_

Round 1

Reviewer 1 Report

This communication is well designed and extensively probed, so it's deserve the publication. 

I have got some comments:

  1. It is not worth mentioning abbreviations (such as  D4 and C2c) in the abstract for which no decoding is given.
  2. Figure 5 is outside the scope of this page.

Reviewer 2 Report

This manuscript describes the oxidation behavior of stainless-steel grade AISI 441 and polymer-derived ceramic (PDC) protective coatings at different temperatures (900, 950, 1000 °C) and exposure times (24, 96 h) in synthetic air and in air saturated with water vapor. The manuscript is overall well prepared and written. I only have a few minor comments below:

  1. I think that the current manuscript can fit into the Special Issue of “Recent Advances in Polymer-Derived Ceramics and Ceramic Nanocomposites” in the current Journal, “Molecules”. However, it can fit into Molecules’ sister Journal in MDPI, “Coatings”, as well.
  2. 3: it is good to show comparative “Scratch morphology in the coating C2c” “after pyrolysis in ambient air at 850 °C/1h”
  3. 4, top right figure: it is good to use double arrows to indicate “bond coat” instead of 1 arrow;
  4. 5: the right most mappings in both rows are missing the elemental labeling (probably Si or Y).

Reviewer 3 Report

The revised manuscript describes the process of fabrication, characterization, and oxidation behavior of composite coatings on commercially available ferritic stainless steel. The problem raised is an interesting issue and the results have a potential application possibility. In the work, the Authors used commercially available precursors, glass, etc. After the preparation of the coatings, the materials were subject to oxidation tests and characterization like scratch tests, microstructure analysis (SEM, EDX), XRD. Generally, the obtained results are discussed accurately, and the manuscript is written correctly. The conclusions are justified. In my opinion, the manuscript can be published after consideration the following remarks

  1. More detailed information concerning the precursor materials should be given. It is especially important in the case of the glass frit. Although, it is a commercially available material some basic information concerning the chemical composition and selected properties should be included. Based on the conducted research and results one can estimate the composition but the parameter may strongly influence the results. In this way, it limits the scientific sound of the paper.
  2. What is the uncertainty of the results? How many samples were measured? Is the data the result of one or several tests?
